# Microencapsulation of Essential Oils and Oleoresins: Applications in Food Products

**DOI:** 10.3390/foods13233873

**Published:** 2024-11-29

**Authors:** Beatriz Fernandes, M. Conceição Oliveira, Ana C. Marques, Rui Galhano dos Santos, Carmo Serrano

**Affiliations:** 1Instituto Nacional de Investigação Agrária e Veterinária (INIAV), Av. da República, Quinta do Marquês, 2780-157 Oeiras, Portugal; beatriz.c.fernandes@tecnico.ulisboa.pt; 2CERENA, DEQ, Instituto Superior Técnico (IST), University of Lisbon, Av. Rovisco Pais, No. 1, 1049-001 Lisbon, Portugal; ana.marques@tecnico.ulisboa.pt (A.C.M.); rui.galhano@tecnico.ulisboa.pt (R.G.d.S.); 3Centro de Química Estrutural, Institute of Molecular Sciences, Instituto Superior Técnico (IST), University of Lisbon, Av. Rovisco Pais, No. 1, 1049-001 Lisbon, Portugal; conceicao.oliveira@tecnico.ulisboa.pt; 4Linking Landscape, Environment, Agriculture and Food-Research Center (LEAF), Instituto Superior de Agronomia, University of Lisbon, Tapada da Ajuda, 1349-017 Lisbon, Portugal

**Keywords:** microencapsulation, essential oils, oleoresins, food products, beverages, food industry, flavorings, preservatives, colorants, functional foods

## Abstract

Essential oils (EOs) and oleoresins (ORs) are plant-derived extracts that contain both volatile and non-volatile compounds used for flavoring, coloring, and preservation. In the food industry, they are increasingly used to replace synthetic additives, aligning with consumer demand for natural ingredients, by substituting artificial flavors, colorants, and preservatives. Microcapsules can be added to a vast range of foods and beverages, including bakery products, candies, meat products, and sauces, as well as active food packages. However, incorporating EOs and ORs into foods and beverages can be difficult due to their hydrophobic nature and poor stability when exposed to light, oxygen, moisture, and temperature. Microencapsulation techniques address these challenges by enhancing their stability during storage, protecting sensitive molecules from reacting in the food matrix, providing controlled release of the core ingredient, and improving dispersion in the medium. There is a lack of articles that research, develop, and optimize formulations of microencapsulated EOs and ORs to be incorporated into food products. Microencapsulated ORs are overlooked by the food industry, whilst presenting great potential as natural and more stable alternatives to synthetic flavors, colorants, and preservatives than the pure extract. This review explores the more common microencapsulation methods of EOs and ORs employed in the food industry, with spray drying being the most widely used at an industrial scale. New emerging techniques are explored, with a special focus on spray drying-based technologies. Categories of wall materials and encapsulated ingredients are presented, and their applications in the food and beverage industry are listed.

## 1. Introduction

Essential oils (EOs) and oleoresins (ORs) extracted from plants, herbs, and spices are already widely used in the food industry as natural food additives, primarily for flavorings in food products and beverages. In addition to flavor, these ingredients offer valuable preservation properties, including antioxidant, antimicrobial, and antifungal effects [1]. With consumers increasingly shifting away from synthetic ingredients in favor of healthier options, there is a growing demand for products free from artificial additives, often identified by “E numbers” on labels. The toxicity of food additives, mainly food dyes, has been re-evaluated by the European Food Safety Authority (EFSA) in the last decades, assessing their acceptable daily intake (ADI) values [2]. Food colorants such as E143 (FD&C green no. 3) and E121 (citrus red no. 2) are banned in the European Union (EU), while being authorized in the US under some restrictions [3,4]. Titanium dioxide (E171), a white pigment previously used in the EU in food products, was re-evaluated in 2021 by an EFSA expert panel to assess the safety of this food additive. Based on results from this evaluation, E171 was considered no longer safe when used as a food additive and removed from the list of authorized food additives [5].

However, incorporating these natural ingredients into food and beverages poses some challenges. Pure EOs and ORs are often unstable within the food matrix due to chemical interactions, as they are prone to evaporation, degradation, and oxidation during storage. High processing temperatures can also destroy their sensitive bioactive compounds. Moreover, their hydrophobic nature makes it difficult to integrate them into hydrophilic matrices [6,7]. Interactions between bioactive molecules and food components can hinder their release properties and antioxidant activity. Interaction with proteins can alter the retention and release of volatile flavor compounds, while fats may prevent the release of components with antimicrobial activity to the water phase of the product, where bacteria grow. The release mechanisms of EO and OR constituents can also be influenced by interactions with carbohydrates by altering the diffusion of volatile molecules within the food matrix or binding through molecular interactions [8].

Encapsulation, typically using polymeric materials, helps in addressing these limitations by protecting the active compounds during processing and enabling controlled release during storage, thus extending shelf life [9,10,11]. Additionally, the strong aromas of pure EOs and ORs can sometimes alter a product’s flavor profile desirability. Microencapsulation solves this issue by masking the flavor and aroma while preserving their bioactive properties [12]. When synthetic preservatives are substituted with microencapsulates, the taste and aroma of EOs and ORs may not align with the desired sensory profile of the product. Selecting appropriate wall materials that effectively reduce the volatility of these compounds while enabling controlled and gradual release over time ensures long-term preservation without significantly compromising consumer acceptability [13]. This approach is similarly applicable to functional foods, where bioactive molecules can remain intact during digestion and subsequently released for absorption in the intended target site [14].

Microencapsulation is a technique widely used across various purposes, such as preserving aromas, masking the taste of bitter components, and protecting ingredients from degradation in formulations [15,16]. It is particularly prevalent in the food, pharmaceutical, medicinal, and cosmetic sectors, where it helps enhance product quality [17]. Bakery products, chocolates, candies, gums, meat products, sauces, and beverages are some of the main products enriched with essential oils in Europe [18]. Formulations of microencapsulated oil products have been developed and patented to be used in sport drinks, milk-based powdered drink mixes, infant formulas, sauces and gravies, baked goods, and snack bars [19]. EOs and microencapsulated EOs have also been used in active food packaging and biodegradable films as antibacterial agents. Their effectiveness has been demonstrated in various studies across a range of food products (e.g., encapsulation of clove bud and oregano EO in sliced-bread packaging [20], bergamot EO in grapes [21], ginger EO in almonds [22], and oregano and pimento EO in meat [23]). The core goal of microencapsulation is to protect the active substance by surrounding it with a continuous polymeric layer. This outer layer, known as the wall, can be composed of a single material or a blend of different materials [16].

Microcapsule walls can be single or multilayered, with the latter offering greater protection, enhanced storage stability, and improved control over the release of the encapsulated compound. Multilayered microcapsules are often produced by incorporating different wall materials in stages or by creating multilayer emulsions, which typically involve adding an emulsifier with the polymer at different stages of emulsion preparation [24,25,26]. A simple scheme of this procedure can be observed in Figure 1. Depending on the technology and materials used, microcapsules can range in size from 0.02 to 10 000 μm in diameter [27,28].

These capsules provide superior protection against lipid oxidation, a critical factor that impacts the shelf life and flavor quality of food products [29]. The increased wall thickness in multilayer capsules enhances the stability of the encapsulated agents and allows for more precise control over the release of the active compounds, as the thickness and composition of the layers can be tailored to release the contents under specific conditions [30].

In the food industry, microencapsulation is employed to incorporate ingredients that otherwise degrade during processing, oxidize in storage, or impart unpleasant tastes or aromas. EOs, ORs, fatty acids, prebiotics, probiotics, antioxidants, and micronutrients are among the ingredients that food manufacturers focus on encapsulating for their benefits and positive impact on the final product [31].

This review paper aims to present various encapsulation methods of EOs and ORs for applications in the food industry. Various relevant topics are mentioned and explored, including common wall materials, encapsulated ingredients and their application in food products, and new emergent technologies for the encapsulation of sensitive compounds. This article explores the potential applications of EOs, ORs, and their microencapsulated forms in the food industry. The objective is to encourage the development of safer, and flavorful food products and beverages that offer additional health benefits to the consumer.

## 2. Microencapsulation Techniques

The encapsulation of active agents can be achieved using various methods, each of which influences the selection of wall materials, the compatibility of the core ingredient, and the resulting capsule’s morphology and characteristics (such as size, release mechanism, encapsulation efficiency, and solubility). These techniques are generally categorized into three types: physical (e.g., spray drying, freeze drying, spray chilling, extrusion, fluidized bed coating, supercritical fluids, and emulsification), chemical (e.g., interfacial and in situ polymerization), and physicochemical (e.g., coacervation, ionic gelation, and liposomes) [30,32,33,34]. Physical methods are usually temperature dependent for capsule obtention, whilst chemical and physicochemical do not rely on high or freezing temperatures for the formation of polymeric walls [32]. From an industry perspective, temperature control can increase costs due to energy demand; however, physical-based technologies are easily scalable, reducing overall price per unit of finished product. Raw materials used in these techniques also present lower cost when compared to chemical and physicochemical methods [35]. Scalability to an industrial level is a challenge for chemical and physicochemical, as their processes are more complex, some may require solvents with higher toxicity levels, and product yield is lower [36].

In a 2015 report by the Business Innovation Observatory from Europe, microencapsulation is presented as a trend with highly economic potential, as consumer demand for healthier food is increasing. This and other novel food-processing technologies help safeguard the environment by using less water and energy during product processing and storage than current technologies. The protection of sensitive compounds by microencapsulation allows for higher shelf-life stability, resulting in less energy spending in refrigeration and product wastage [37].

In the food industry, EOs and ORs are commonly encapsulated through methods like spray drying, freeze drying, in situ polymerization, or fluidized bed coating, though other established and emerging techniques have also been used [38,39]. However, while some methodologies developed in laboratories show promising results, scaling them up for industrial production can be challenging. These challenges often include the complexity of equipment, high processing costs, and the impracticality of using such methods on a large scale, making them less viable for commercial production. [40,41].

Table 1 provides a collection of studies since 2013 that examine microencapsulation of EOs and ORs for food applications, using different techniques and highlighting the role of microcapsules in product formulations.

Spray drying is the most used technique at an industrial scale for producing microencapsulated powders of bioactives, flavors, oils, enzymes, and others. Its popularity in the food sector is driven by three key factors: low operational cost, high production capacity and process simplicity [30]. A variety of food-grade materials with diverse encapsulating, film-forming, and emulsifying properties can be used in spray drying, offering manufactures choice in selecting ingredients that meet both desired product characteristics and budget constraints [57].

Briefly, spray drying consists of four steps, namely preparation of the feed solution, atomization of the prepared solution, evaporation of moisture, and separation of the obtained powders [58]. The conditions under which these operations are performed play a crucial role in the production, as well as the quality of the final product [59]. The feed solution for microencapsulated powders comprises an emulsion, solution, or suspension containing the core material and encapsulating agents [60]. The homogenization process greatly influences droplet size and size distribution of the atomized capsules, impacting the powder’s physicochemical characteristics, final yield, and shelf-life stability [58]. The feed solution is pumped onto the atomizer head and sprayed, where it comes in contact with a stream of hot air, the drying medium. In a few seconds, the water is evaporated inside the drying chamber, and the resulting powders are collected through a cyclone separator and into a vessel [58,61]. During this process, three variables can determine the powders characteristics: feed, mainly flow rate and concentration; drying agent supply, including air flow rate, inlet temperature, outlet temperature, and humidity; and selection of atomizer, regarding the type, diameter and number of nozzles, and rotational velocity [62]. A simple schematic figure of the spray dryer is shown in Figure 2.

An analysis of published articles since 2010, conducted through the Scopus database, examined research trends across various microencapsulation techniques. The terms “microencapsulation” and “food” were combined with specific techniques such “spray chilling”, “liposomes”, “fluid bed coating” (or “fluidized bed coating”), “polymerization”, “coacervation”, “freeze drying”, and “spray drying”. Figure 3 shows that spray drying is the most extensively researched microencapsulation technique in the food industry.

As the primary microencapsulation method across various industries, optimizing the spray-drying process is critical for producing microcapsules with desirable physicochemical characteristics. Many studies focus on optimizing key factors such as wall-material selection, blending ratios, and atomization parameters to achieve high encapsulation efficiencies, appropriate release mechanisms, and optimal product quality. The ideal parameters depend on the material being encapsulated and its intended application [30].

One study explored the optimization of a cost-effective spray-dried microencapsulated EO using gum arabic, maltodextrin, and modified starch [44].

The researchers tested thirteen formulations: three using single wall materials, six with binary blends, and four with ternary blends. Inlet temperatures (150, 160, 175, and 180 °C) and oil concentrations (10, 20, and 30%) were also tested. The results showed that a drying temperature of 175 °C and a 10% oil loading resulted in the highest encapsulation efficiency (77.50%). Among the formulations, a 50:50 blend of gum arabic and maltodextrin had the highest efficiency (83.60%), while 100% gum arabic yielded the best results among single agents (77.50%). For ternary blends, a mix of one-third gum arabic, one-third maltodextrin, and one-third modified starch resulted in a slightly higher efficiency (77.79%).

Another study investigated the optimization of lavender EO microcapsules, evaluating the effects of total solids concentrations, oil loadings, and gum arabic ratios in 27 formulations [63]. The authors used maltodextrin and gum arabic as wall materials and tested three levels of solids (25, 30, and 35 *w*/*w*%), oil-to-total-solids ratios (16.67, 20, and 25 *w*/*w*%), and gum arabic concentrations (25, 40, and 50 *w*/*w*%). With a fixed inlet temperature of 150 °C, the highest encapsulation efficiency (77.89%) was achieved with the lowest oil loading (16.67%), the lowest gum arabic concentration (25%), and a solids concentration of 30%.

Similarly, Nhan et al. [64] optimized microencapsulation of lemon EO using spray drying. They tested wall-material concentrations (15–30%), wall-material types (maltodextrin and gum arabic, in single or binary blends), and EO concentrations (0.5–2%). Higher microencapsulation yields and efficiencies were observed with 30% wall materials, either 100% maltodextrin or a binary blend of maltodextrin and gum arabic. However, 100% maltodextrin showed slightly better results. The encapsulation yield was highest at 1.5% oil, while the encapsulation efficiency peaked at 1%. The efficiency decreased significantly at the highest oil concentration (2%).

In another study [65], researchers evaluated spray-dried encapsulated ginger EO by varying inlet temperatures (140, 155, and 170 °C) and solids concentrations (20, 25, and 30%). Whey protein isolate and inulin were used as encapsulating agents, with an oil-to-wall-material ratio of 25 *w*/*w*% across all trials. Contrary to other studies, the highest encapsulation efficiency occurred at a lower solids concentration (20%) and a higher drying temperature (170 °C). Using response surface methodology, the optimal conditions were determined to be 22.34% whey protein isolate to inulin and an inlet temperature of 170 °C.

Comparing encapsulation efficiency across different studies is challenging due to variations in methodologies and terminology, with different researchers using terms like “encapsulation efficiency”, “entrapment efficiency”, and “loading efficiency” interchangeably. As the results suggest, optimal encapsulation parameters depend on multiple factors, including the choice of wall materials, their ratios, core- and wall-material concentrations, air-drying temperatures, and the composition of the core material.

### Emerging Encapsulation Technology

Most of the microencapsulation technologies discussed in this section have been extensively studied and are widely used in the food and pharmaceutical industries. New encapsulation methods, primarily developed at the laboratory scale, focus on improving the stability of sensitive compounds, exploring novel wall materials with unique properties, and facilitating the upscaling of processes at lower costs. Advances in spray-drying techniques, in particular, have addressed some of the limitations of traditional methods. Emerging technologies such as nano spray drying, vacuum spray drying, ultrasound-assisted spray drying, spray freeze drying, dehumidified-air spray drying, and microfluidic-jet spray drying offer alternatives to conventional spray drying. These were developed as a way of circumventing the difficulties and disadvantages found with the currently used technologies, such as low powder yield, necessity of high processing temperatures, heterogeneous particle sizes, and subpar physicochemical qualities [57]. These methods, summarized in Table 2, highlight key differences in instrumentation, advantages, and limitations compared to regular spray drying.

Original batch-based processes like fluid bed coating have been modified for continuous operation, reducing costs and increasing production capacity [66]. Although these emerging encapsulation methods show promise for the food industry, further research is needed to reduce production costs, streamline scale-up procedures, identify suitable encapsulating agents for different core ingredients, and optimize formulations for specific applications [57].

The high initial costs of scaling up production can be a barrier for food industries when adopting new technologies and equipment. While innovative microcapsules may be more expensive, they can offer superior benefits in terms of product characteristics—whether in regard to flavor, preservation, or health attributes. These emerging encapsulation techniques may find greater success in niche markets, such as functional foods, as consumers become increasingly focused on health-conscious choices [28].

Spray-drying technology began to be applied in the industry in the 1920s for the production of dried milk powder, subsequently evolving to serve a range of industries, including food, chemical, and pharmaceutical sectors [67]. Over the past century, advancements in spray-drying processes and equipment have been continuously implemented to address industrial demands and maintain competitiveness in the market. Notable innovations in dryer design include the integration of magnetic rotary atomizers, integrated filters, and multistage spray drying combined with fluidized bed systems [68]. Nano spray drying, a more recent advancement, enables the production of nanocapsules characterized by enhanced solubility, improved bioavailability, higher product yields, and superior encapsulation of thermally sensitive ingredients. Despite these benefits, the scale-up of nano spray drying remains a significant challenge. Key limitations include atomizer head blockages caused by viscous feed solutions, extended processing times, and low product throughput [69]. The market for nanoencapsulation is experiencing significant growth, with companies increasingly investing and promoting projects that leverage this innovative technology. Several commercially available products incorporate nanoencapsulated ingredients in their formulations. Examples include “Daily Boost”, a beverage enriched with nanoencapsulated vitamins and bioactive compounds developed by Jamba Juice (Georgia, USA); “Tip Top Up Omega-3 DHA”, a fortified bread containing nanoencapsulated omega-3 DHA-rich fish oil produced by Tip Top (Delhi, India); and “Nanoceuticals™ Slim Shake Chocolate”, a low-fat drink formulated with CocoaClusters™, nanoencapsulated cocoa to enhance flavor without the need for added sugar, manufactured by RBC Life Sciences^®^ (Texas, USA) [70,71,72,73].

The upscale potential of some of these new technologies could be supported by changes in legislation when it comes to incorporation of nanomaterials in the food industry, as they are categorized as novel foods, and their authorization is more difficult to obtain. Low production yields are often due to lack of formulation optimization for the specific core ingredient being encapsulated. Research on a laboratory scale can be useful for the development of better formulations and improvement of processes, resulting in cost reduction in the industry [74].

**Table 2 foods-13-03873-t002:** Emerging encapsulation technologies derived from spray drying.

Technology	Differences	Advantages	Disadvantages	Scale-Up Challenges	References
Nano spray drying	Nebulizer is used as nozzle; air flow is laminar (SD is turbulent); electrostatic particle collector	Smaller particles, in the nano range (10 nm to 1 μm); higher bioavailability and more precise controlled release	Harder to scale up production; nanomaterials follow different EU regulations for food applications	Nanomaterial regulations	[75,76,77]
Vacuum spray drying	Drying chamber, under reduced pressure; superheated steam is used as a heat source	Low drying temperatures (30–60 °C); absence of air flow reduces oxidation; continuous process; higher EE	Lacking research in the encapsulation of EOs and ORs; higher costs	High production costs	[61,78]
Ultrasound-assisted spray drying	Spray droplets are formed with ultrasonic nozzle atomizers; high energy vibration induces droplet formation	Small and uniform capsules; mechanical stress is low, better for bioactive molecules	Choice of wall materials is limited; lack of studies on encapsulation	Low production yield	[61,79,80,81]
Spray freeze drying	Feed solution is atomized, frozen with cold dry gas or liquid, and sublimed at low temperatures and pressure	Faster process than FD; lower temperatures than SP; drying conditions can be altered; spherical particles with better oxidative stability; continuous approach is under development	Research in encapsulation is lacking; high production costs and harder to scale-up; core ingredient could affect freezing process	Semicontinuous process, high production costs	[82,83,84,85]
Dehumidified air spray drying	Dehumidified air is used for drying; spray-dryer air intake is connected to a dehumidifier unit	Lower drying temperature is required; less powder stickiness leads to better powder recovery; particles show lower moisture content	Lacking research on encapsulation effects	Minor, easy implementation to spray dryer system	[86,87]
Microfluidic-jet spray drying	Atomizer produces monodisperse droplets as a result of its nozzle (SD is polydisperse, droplets can collide and agglomerate)	Droplets are uniform in size and spherical in shape; less variation in size; reduces aggregation; easier to simulate process through mathematical models; higher EE	Nozzle may block when using larger particles in feed solution; harder to scale up due to the small capacity of the nozzle	Low production yield	[88,89]
Electrospraying	Droplets are formed with a controlled supply of electrostatic fields through an electrically conductive nozzle	Particles smaller than 1 μm; cost effective and potential to be scaled up; low drying temperatures	Low production volume; amounts of wall materials are very limited; nanomaterials follow different EU regulations for food applications	Low production yield	[90,91,92,93]

EE, encapsulation efficiency; EOs, essential oils; FD, freeze drying; ORs, oleoresins; SD, spray drying.

## 3. Encapsulation Materials

The selection of coating materials plays a crucial role in determining the structure and properties of the resulting microcapsules. It requires careful consideration of the core material’s chemical nature, the drying method used, the purpose of the encapsulated ingredient, and the matrix in which it will be applied. In addition to primary encapsulating agents, the trigger for its release, or the requirement for its enclosure and protection, other components, such as antioxidants, surfactants, and emulsifiers, can be incorporated into the coating layer to enhance functionality [94]. The effects of emulsifiers in spray-dried microcapsules of Wilson’s dogwood (*Swida wilsoniana*) oil were explored by Yang et al. [95] in a 2020 paper. The properties of lecithin, tween 80, and their blends were characterized by their emulsion size, creaming stability, encapsulation efficiency, and oil loading. The results showed the importance of emulsifier choice in the production of cost-efficient and stable microcapsules, as it is highly relevant at an industrial level.

The nature of the wall material influences the release mechanisms of the intact capsules, and their solubility and permeability should be considered for the intended application. A US patent from 2005 [96] developed a system to improve the controlled release of encapsulated flavors, sensory markers, and active ingredients for applications in food products, beverages, nutraceuticals, and oral care products. The invention is designed to encapsulate single or multiple active ingredients with multiple layers of distinct wall materials, both hydrophobic and hydrophilic. This allows for the release of different active ingredients when exposed to different environments. The authors mention the triggered release of flavors encapsulated in water-sensitive wall materials upon exposure to moisture (e.g., during ingestion) and a slower release of the agent protected by the hydrophobic material over an extended period of time. Hydrophobic wall materials mentioned in the patent include carnauba and candelilla wax, while the list of hydrophilic materials comprises water-soluble and -dispersible synthetic polymers, starch derivatives, gums, polysaccharides, proteins, and hydrocolloids.

The choice of microencapsulation technique directly influences the selection of encapsulating agents, and vice versa. Since the capsules are intended for consumption, the materials must be classified as Generally Recognized as Safe” (GRAS) and approved by regulatory bodies like the EFSA and FDA [97]. Additionally, the choice of encapsulating materials depends on the final food product and the intended function of the microcapsules within the formulation. Approved wall materials in the food industry can be plant-based, marine-based, animal-based, or microbial-based [98].

Goud and Park [99] categorize wall materials based on their molecular composition—such as carbohydrates, cellulose, gums, lipids, and proteins. Carbohydrate polymers, particularly starches and gums, are the most commonly used wall materials in the food industry due to their low cost and wide applicability [34,100]. Wall materials can be classified in several ways, including by chemical composition (e.g., carbohydrates, cellulose, gums, proteins, and lipids) [99], origin (e.g., plant-based, animal-based, microbial-based, and marine-based) [101], polymeric or non-polymeric structure, or solubility (water-soluble or water-insoluble) [102].

In addition to the primary encapsulating agents, which form a protective film around the active ingredient, other compounds can be added to the wall layer to enhance protection of the core material and control its release during storage or consumption. Emulsifiers and antioxidants are the most commonly used additives in food-industry formulations [98,103]. In many microencapsulation techniques, the core ingredient must be dispersed into a solution containing hydrated wall materials, a process that often involves emulsification. The size of the resulting droplets can be controlled, as particle size significantly influences key properties, such as solubility, oxidation rate, and release profile of the core ingredient [98]. Some encapsulating agents, like gum arabic, proteins, and modified starches, possess both emulsifying and film-forming properties, making them popular choices for food applications [62].

From an industrial perspective, two important parameters are wettability and solubility, as they determine how well the powders integrate into a product. Wettability, which reflects how easily powders absorb water, is measured in time units such as min/g or seconds—the lower the value, the faster the absorption [104]. Solubility refers to how much of a substance dissolves in a solvent and is mainly influenced by the type of wall material used [105]. Proper selection is essential, as poor solubility can lead to difficulties during food or beverage processing [106]. Starch-based materials are favored for their good water solubility, with maltodextrin being a common choice due to its low cost, neutral flavor, oxidative protection, and low viscosity at high concentrations [107]. Maltodextrins are categorized by their dextrose equivalent (DE), which correlates to the length of glucose chains; a higher DE results in shorter chains, giving the powder a sweeter taste and increased solubility, as demonstrated in Goula and Adamopoulos’ paper [108].

Processing conditions, like inlet air temperature during spray drying, can also impact wettability and solubility. For example, increasing drying temperatures from 150 °C to 190 °C significantly improved the solubility of microcapsules made from 100% gum arabic, while reducing wettability in microcapsules made from maltodextrin, gum arabic, or 100% inulin [106]. However, some studies, such as one by Fernandes et al. [107], which encapsulated rosemary EO using different ratios of gum arabic, modified starch, maltodextrin, and inulin, found no significant differences in solubility. Wettability, however, was affected by wall composition, with inulin significantly reducing wettability time.

## 4. Core Materials: Essential Oils and Oleoresins

A significant portion of encapsulated active agents in the food industry comprises EOs and ORs, valued for their flavor and bioactive properties [109]. The advantages of these microcapsules also include enhanced color stability and additional nutritional benefits. In the food sector, EOs and their microcapsules are utilized for their flavor, aroma, antioxidant properties, and stability, serving as effective food preservatives [110].

EOs and ORs are natural extracts obtained from plant-based materials, comprising volatile components in the case of EOs, whereas a majority of OR’s components are non-volatile. EOs are often extracted through steam distillation processes, hence only removing the volatile fraction of the plant [111]. During extraction of ORs, the plant material is in direct contact with the solvent, usually through Soxhlet extraction method. This allows for the transfer of the non-volatile compounds from the plant to the solvent, which is after evaporated and concentrated to obtain a thick, viscous resin [112,113]. These extracts are less susceptible to degradation from higher temperatures than EOs, since the resinous fraction acts as a fixative [113]. However, exposure to light and oxygen causes both extracts to deteriorate [6,114].

ORs not only contain the volatile compounds found in EOs but also include non-volatile constituents, resulting in a more complex composition, flavor, and color. They demonstrate higher heat resistance and a more stable aroma, as the resin portion acts as a fixative, protecting the volatile EO components [115]. Encapsulating both extracts helps safeguard them during processing and storage from high temperatures and oxidation, thereby preserving their flavor, color, and bioactive properties [27].

The microcapsules can be more easily dispersed in aqueous environments, addressing issues related to the poor water solubility of EOs [116]. Sphera Encapsulation, an Italian start-up, used micro- and nanoencapsulation methodologies to develop SpherAQ^®^, an unique technology that encapsulates lipophilic ingredients, protecting them from degradation and increasing solubility in water, enabling the incorporation of insoluble ingredients in beverages and foods [117,118]. An example of a commercial product that uses encapsulated EOs and ORs in their formulation is the line EZ-Caps™, designed to be used in beverages, instant drink mixes, dry seasoning, and sauces [119]. For this type of application, the capsules with hydrophilic wall materials release the core ingredients mainly by diffusion, by forming a hydrated gel layer [102].

Solubility and dispersibility of the encapsulated powders are important factors in the food industry, especially in the beverage sector. The formation of powder agglomerates after addition to the liquid beverage negatively impacts consumers perception of the product [120]. Encapsulation of water-insoluble materials with appropriate encapsulating agents (mainly polysaccharides) increases solubility and dispersibility in water, allowing the use of plant extracts in beverages or food products with high water content [121,122].

However, one challenge in using EOs and ORs in food products is the maximum allowed concentration in formulations [123]. Standardizing the safety of EOs for consumption can be complex due to the existence of approximately three hundred different types, each with its own origin, usage, and composition [124]. In the United States, the FDA has classified over 150 types of EOs and ORs as Generally Recognized as Safe (GRAS) [125]. In the European Union, EOs and ORs listed among usable plant species can be classified as flavorings since they are 100% natural extracts derived from plant materials, in accordance with European Regulation 1334/2008 [126].

The EFSA assesses the safety of EOs for consumption based on the Threshold of Toxicological Concern (TTC), establishing safe oral doses for each component of the EO under study [127,128]. Overall, the incorporation of EOs and ORs into food and beverages intended for human consumption depends on the specific ingredient, application method, and intended purpose, as well as the regulatory environment of the country where the product will be sold [129].

## 5. Food and Beverages Applications

EOs and ORs are renowned for their strong aromas, attributed to the volatile compounds present in these extracts. Microencapsulation of these ingredients protects their taste and aroma from evaporation and oxidation during processing and storage [62]. Many EOs and ORs contain biological active compounds that have been studied for their antioxidant, antimicrobial, antifungal, and preservative properties in food, beverages, and packaging [130].

Encapsulating these extracts also safeguards their sensitive compounds from oxidation within the capsule while slowing down lipid and protein oxidation in the food products they are added to [131]. The capsules can be engineered to burst and release their active agents at specific times and under certain conditions. This release mechanism is influenced by the microcapsule’s composition, size, geometry, and the environment of the surrounding matrix. In the food industry, this method can be applied in systems that benefit from the controlled release of specific ingredients at designated pH levels or temperatures [132].

The controlled release mechanisms primarily occur through melting, degradation, rupture of the capsule, or diffusion of the core ingredient through the wall material [132]. The choice of wall materials and microencapsulation techniques is crucial for developing powders with effective controlled-release properties. For example, spray drying typically produces water-soluble powders that quickly release their core agents into a hydrophilic matrix [133]. In contrast, coacervation can yield microcapsules that are insoluble in water, allowing for a more controlled release of their core materials [134].

The selection of wall materials plays a major role in the microcapsules’ flavor retention, release, and stability. It is not an easy process, as the effects of wall-material composition depend on the technology and parameters used, concentrations and ratios between encapsulating agents, and the nature of the substance being encapsulated [135]. Carbohydrates are the most used for encapsulation in the industry, as they are hydrophilic, they cheap, and they have great film-forming and barrier properties [66]. An analysis of microencapsulation of oils in food-product patents demonstrated maltodextrin and modified starch to be the most used wall materials, followed by gum arabic, β-cyclodextrin, fish gelatine, and isolated soy protein. Only the two last mentioned materials are not carbohydrates, demonstrating the significance of carbohydrate-based encapsulation agents in the food industry [136].

Maltodextrin’s main advantages are its low cost; neutral flavor; high protection against oxidation; and low viscosity at high solid concentration, a useful characteristic for spray-drying processes, as it allows for a shorter drying time. However, maltodextrin does not have good emulsifier properties or good volatile retention. Gum arabic can be mixed with maltodextrin to complement its disadvantages, as it has excellent emulsifying and film-forming properties and good retention of volatiles, as well as low viscosity at high concentrations and high oxidative stability. The main disadvantage is its high price; hence, the industry is looking for cheaper alternatives with similar encapsulating characteristics [107,137]. β-cyclodextrin is an inexpensive polymer with great properties for the entrapment of lipidic ingredients, as this molecule has a cage-like structure, hydrophobic on the inside and hydrophilic on the outside [138]. Encapsulation is often performed via inclusion complexation, in which the size of its cavity influences the loading capacity of the entrapped material [98,135]. Although β-cyclodextrin usage is not limited in the USA, it is a food additive (E459) in the EU with an ADI of 5 mg/kg of body weight per day [139], and it is only authorized to be added to a limited number of products [140].

When selecting the appropriate technique and materials, it is essential to consider the matrix of the food product. The following subsections will explore the main applications of microencapsulated EOs and ORs in the food sector, highlight relevant articles, and discuss potential uses of these powders in food and beverages.

### 5.1. Preservative

EOs and ORs are concentrated extracts extensively utilized in the food industry across various sectors, including confectionery, bakery, dairy products, and beverages [111]. They are incorporated into recipes primarily for preservation purposes. Growing consumer concerns about natural food additives have prompted the industry to replace synthetic ingredients with natural alternatives derived from plants, animals, fungi, and algae [141]. EOs and ORs are known to possess antimicrobial, antifungal, antioxidant, and anticancer properties [142,143,144].

While herbs and spices, whether powdered or fresh, are often used for flavoring in food, they are typically present in insufficient quantities to exert a significant preservative effect. Additionally, they may be contaminated with pathogens and heavy metals, posing potential public health risks [145,146]. Plant extracts, on the other hand, offer a safer means of incorporating bioactive compounds into food and beverages, as they are of standardized quality and free from microbial or other contaminants [147]. Moreover, their concentrated nature allows for the use of smaller quantities to achieve the desired effects.

The application of non-encapsulated EOs and ORs in food systems and their preservation effects has been well studied, not only in processed foods but also in fresh fruits and vegetables [127]. Despite the abundance of research on pure EOs and ORs, studies focusing on encapsulated extracts and their incorporation into food and beverages remain limited. Nonetheless, research articles have explored the antioxidant and antibacterial effects of these extracts in greater detail compared to their other common uses, which will be discussed in the following subsections.

Several studies have addressed the nanoencapsulation of EOs in food products; however, these are not included in this paper since nanomaterials are regulated differently than microcapsules. In the European Union, nanomaterials are classified as novel foods under Regulation 2015/2283 and must undergo safety testing and explicit authorization for use in food products [148]. When nanocapsules are used as food additives, a different set of regulations (Regulation 1333/2008) applies [149]. Changes to approved additives through nanotechnology are treated as new products that require re-evaluation. Additionally, the presence of these nanomaterials must be indicated on labels, which may deter uninformed consumers [150,151].

In contrast, the United States Food and Drug Administration (FDA) does not regulate the use of nanomaterials in food and beverages, instead offering technical advice and guidelines. Food products containing nanocapsules can be marketed without pre-market approval, as they are classified as Generally Recognized as Safe (GRAS) substances [152,153].

In the analysis of published articles evaluating the effectiveness of microencapsulated EOs in food products, three studies used cake as the test medium, with two of them authored by the same lead researcher. In one study, orange EO was microencapsulated using β-cyclodextrin through precipitation, followed by anti-solvent precipitation with zein and electrospraying to obtain the powder form [154]. In vitro microbiological tests showed that free essential oils (FEOs) exhibited antifungal activity against various *Aspergillus* species, with minimum inhibitory concentration (MIC) values ranging from 45.24 to 90.47 mg/mL and minimum fungicidal concentration (MFC) values ranging from 45.25 to >180.95 mg/mL. Higher MIC and MFC values indicate that a greater quantity of a product is required to inhibit microorganism growth, making it less effective as an antimicrobial or antifungal agent.

An in situ analysis demonstrated that both FEO and encapsulated essential oil (EEO) delayed fungal growth in cake samples compared to the control. By day 30 of storage, the control cake had a mold and yeast count (MYC) of 2.5 × 10^3^ ± 0.01 CFU/g, whereas no fungal growth was observed in the FEO and EEO samples. Fungal colonies in the FEO and EEO cakes began to appear around day 150, with MYC values of 1.1 × 10^3^ ± 0.01 and 1.4 × 10^3^ ± 0.01 CFU/g, respectively, while the control had a significantly higher count of 26.4 × 10^3^ ± 0.02 CFU/g [154].

Thermal resistance was also analyzed, as orange FEO and EEO were meant to be incorporated in cake batter and baked at high temperatures (180–220 °C). The concentration of nine volatile compounds was measured after FEO and EEO samples were exposed to 180 °C for 20 min to simulate baking conditions. Water activity across all samples, including the control, remained stable at approximately 0.80 from day 0 to day 30, indicating no moisture loss during storage. Limonene, the major volatile compound, was used to compare the samples. A 5% *w*/*w* solution of EEO showed a higher concentration of limonene after heat treatment (625.12 mg/g) compared to FEO (427.46 mg/g), demonstrating that encapsulation provided better thermal protection. This enhanced thermal resistance helped preserve the antifungal properties of the EO during cake preparation [154].

The second study by the same lead author utilized encapsulated orange oil with β-cyclodextrin, produced through co-precipitation and vacuum filtration [155]. In vitro tests showed that FEO exhibited antifungal activity against *Aspergillus flavus*, with a MIC of 0.45 mg/mL and a MFC of 0.9 mg/mL. In contrast, the EEO only demonstrated inhibition at the highest concentration tested (MIC of 900 mg/mL) and showed no MFC, likely because the oil remained trapped inside the microcapsules.

Both FEO and EEO were incorporated into cake batters, baked, and analyzed for fungal growth after 15 days. Unlike the authors’ previous study, the EEO-infused cake did not exhibit significant antifungal activity compared to the control (EEO MYC, 76.0 × 10^3^ ± 4.30 CFU/g; control MYC, 81.5 × 10^3^ ± 5.00 CFU/g). However, the FEO-infused cake was effective, with a lower MYC of 13.5 × 10^3^ ± 3.20 CFU/g. The authors attributed the lack of antifungal activity in the EEO sample to the microcapsules not releasing the entrapped oil during baking. Although the oven temperature reached 220 °C, the internal crumb temperature only reached 98 °C—insufficient to trigger the release of the core material [155].

In a 2017 study, thyme EO was selected as a natural preservative for cake samples and microencapsulated using gelatine and gum arabic through complex coacervation [156]. An in vitro microbiological analysis of FEO demonstrated high activity against eight out of nine tested microorganisms, with MIC values ranging from 0.125 to 0.50 mg/mL. The only exception was *Enterococcus faecium*, which had a higher MIC of 0.60 mg/mL. EEO exhibited even lower MIC values, indicating greater effectiveness against the microorganisms. The MIC values, based on the mass of moist particles used, ranged from 0.14 ± 0.041 to 0.43 ± 0.066 mg/mL, while the MIC based on the oil mass inside the capsules ranged from 0.010 ± 0.002 to 0.030 ± 0.006 mg/mL. This reduction in concentration was attributed to the long-term protection provided by the encapsulating wall materials, resulting in a slow and controlled release of the volatile active compounds.

The preparation of cake samples with FEO and EEO in this study differed from the methods used in previous papers. Instead of incorporating the oils into the cake batter before baking, the batter for all three samples (FEO, EEO, and control) was identical. To evaluate the effect of thyme oil, solutions of FEO (0.125 mg/mL) and EEO (0.125 and 0.600 mg/mL) were sprayed onto the cakes immediately after baking. The results showed that the highest concentration of EEO completely inhibited yeast and mold growth even after 30 days (MYC: <1 log CFU/g). Lower concentrations of EEO and FEO also significantly reduced yeast and mold counts compared to the control after 30 days (EEO, 1.73 log CFU/g; FEO, 1.77 log CFU/g; control, 5.21 log CFU/g). As with previous studies, water activity remained stable over the 30-day period, maintaining a value of approximately 0.87 [156].

A 2020 study tested spray-dried thyme EO microencapsulated with sodium casein and maltodextrin in meat burgers, evaluating its preservative effects [157]. In vitro tests showed that both FEO and EEO exhibited equal MIC and minimum bactericidal concentration (MBC) values (0.1 mg/mL) against *Salmonella typhimurium*, *Listeria monocytogenes*, *Staphylococcus aureus*, and *E. coli*, indicating potent antimicrobial activity for both forms.

For the in situ analysis, five burger formulations were prepared: a control with the standard recipe, a control with capsules but no oil, a burger with sodium nitrite (a commercial preservative), a burger with FEO (0.1 g of thyme EO per 100 g), and a burger with EEO (1 g of microcapsules per 100 g). After 14 days, both controls had the highest counts of thermotolerant coliforms, as expected (*E. coli*, 9.2 /g; thermotolerant coliforms, 460 /g). The burgers with nitrite inhibited the growth of thermotolerant coliforms and *E. coli* after 7 days, with these levels remaining stable at day 14 [157].

When comparing FEO and EEO burgers, the EEO burger demonstrated greater antimicrobial activity against *E. coli*, completely inhibiting growth by day 7. In contrast, the FEO burger showed *E. coli* counts of 3.6 /g on day 7 and 9.2 /g on day 14, indicating no significant inhibition, likely due to the volatilization of the oil’s antimicrobial compounds. However, regarding thermotolerant coliforms, the FEO burger was more effective, reducing the count to 9.2 /g by day 14, whereas the EEO burger had a final count of 15.0 /g [157].

Antioxidant activity was assessed using DPPH, hydroxyl, and nitric oxide assays. FEO exhibited higher antioxidant activity in the DPPH and hydroxyl assays (83.1% and 27.4%, respectively) compared to EEO (5.2%, and not detected), while no significant differences were found in the nitric oxide assay (FEO, 14.8%; EEO, 13.8%) [157].

Thyme EO was also studied for its effects on minimally processed lettuce when encapsulated [158]. The oil was microencapsulated with β-cyclodextrin through co-precipitation, and three lettuce samples were prepared: a control (lettuce with no added oil), FEO-treated lettuce (dipped in an aqueous solution containing 0.5, 1.0, or 1.5 g/L of FEO), and EEO-treated lettuce (dipped in an aqueous solution containing 0.33, 0.66, or 0.99 g of thymol per liter). Microbiological quality was evaluated by tracking the growth of total mesophilic aerobic bacteria, psychrotrophic bacteria, *Enterobacteriaceae*, and yeast and mold over a 12-day storage period.

EEO-treated lettuce samples showed better bacterial growth inhibition overall after 12 days, except for *Enterobacteriaceae*, where only the highest concentration of EEO significantly reduced the count compared to the control by the end of storage. However, bacterial counts in the EEO samples remained significantly lower than the control until day 9. For mesophilic aerobic bacteria, psychrotrophic bacteria, and yeast and mold counts, FEO-treated samples initially exhibited greater bacterial reduction by day 3, but their counts surpassed those of the EEO samples after day 6. This trend aligns with previous studies, where FEO offers stronger immediate protection, but EEO provides sustained bacterial inhibition due to the slow release of active compounds from the microcapsules during prolonged storage [158].

In terms of yeast and mold growth, all EEO concentrations significantly delayed growth over the 12-day period, while FEO-treated samples showed no significant difference from the control. The total phenolic and flavonoid content showed contrasting trends between FEO- and EEO-treated lettuces: FEO samples had higher initial phenolic and flavonoid content, but these levels decreased during storage. Conversely, EEO samples, which started with lower phenolic and flavonoid content, saw an increase over time, ultimately surpassing FEO content across all concentrations by the end of storage. This trend was also observed with DPPH antioxidant activity, further supporting the idea that EEO’s controlled release of bioactive compounds enhances its antimicrobial and antioxidant effects over time [158].

Lettuce weight loss remained consistent among all samples throughout the 12 days of storage. However, FEO-treated lettuce experienced significant color changes, with a noticeable reduction in lightness and green color attributes from day 0 to day 12, while EEO-treated samples showed no significant differences from the control. The application of FEO also negatively affected the organoleptic quality of the lettuce. Panelists reported lower scores for color and texture, as well as a strong aroma, which reduced the overall acceptability of FEO-treated samples. This strong aroma was not detected in EEO-treated lettuce, as encapsulation effectively masked the scent of the EO. By the end of the 12-day period, the only sample with an acceptable overall visual quality (OVQ > 2.5) was the EEO-treated lettuce at the lowest microcapsule concentration [158].

Spray-dried rosemary EO encapsulated in modified starch and maltodextrin was applied to fresh dough to assess its antimicrobial properties [159]. In situ antifungal activity tests were conducted on *Penicillium* sp. and *Aspergillus* sp. fungi using five oil concentrations (1, 5, 10, 20, and 50 μL/mL). All concentrations delayed fungal growth, with the highest concentration (50 μL/mL) completely inhibiting *Penicillium* sp. growth from day 0 to the end of the 20-day assay. For *Aspergillus* sp., concentrations of 20 and 50 μL/mL fully inhibited growth until day 20. Rosemary EO proved effective against both fungi, with *Aspergillus* sp. being more sensitive to its antifungal properties.

For the in situ evaluation, three dough samples were prepared: a control dough, a dough containing 1.5% FEO, and a dough containing 1.5% EEO. Fungal and yeast counts were measured after 4, 8, and 12 days. By the end of the 12-day period, the EEO dough had the lowest fungal and yeast counts (7.53 ± 0.01 log CFU/g), while both the FEO and control samples exhibited higher, uncountable counts (>8.00 log CFU/g) [159].

Consistent with previous studies, FEO samples showed greater initial antifungal activity, recording lower counts (4.10 ± 0.09 log CFU/g) than the EEO samples (5.29 ± 0.01 log CFU/g) on day 4. However, by day 8, the EEO dough had lower fungal and yeast counts compared to all other samples. The encapsulation process gradually released the rosemary oil from its core, protecting the volatile compounds from evaporation and oxidation, which allowed for sustained control of fungal and yeast growth [159].

Moisture content remained consistent across all samples and storage periods, with the exception of the EEO dough on day 0, which had a slightly lower moisture content than both the control and FEO dough samples [159].

Spray-drying technology was also used to test microencapsulated garlic EO as a natural preservative in minced meat, comparing it to freeze-dried and oven-dried garlic [160]. The EO was encapsulated using maltodextrin and gum arabic as wall materials at four concentrations: 5, 10, 15, and 20%. Microbiological analyses were conducted solely in situ, and FEO was not tested in this study.

Total aerobic mesophilic flora counts revealed that only the 20% EEO (4.1 ± 0.3 log CFU/g) and freeze-dried garlic (4.4 ± 0.2 log CFU/g) minced-meat samples remained within satisfactory quality standards after 4 days of storage at 8 °C (counts below the established limit of 5.7 log CFU/g). By day 6, the flora count in the 20% EEO samples increased to 6.4 ± 0.4 log CFU/g, which no longer met the criteria for satisfactory quality but was still deemed acceptable. In contrast, freeze-dried garlic continued to exhibit stronger antimicrobial effects, with a total count of 5.0 ± 0.1 log CFU/g at day 6 [160].

Regarding the effect of different EEO concentrations, only the highest concentration (20%) followed the trend seen in other studies, where increased antibacterial activity was observed as bacterial growth was reduced during the trial. *E. coli* counts showed that all EEO concentrations effectively inhibited the growth of this pathogen after 6 days of storage (<1 log CFU/g). The same result was observed for sulfite-reducing anaerobes, with all EEO concentrations maintaining counts below 1 log CFU/g by day 6 [160].

Although freeze drying is not commonly used for microencapsulation in the food industry, it was employed to encapsulate Sichuan pepper EO using soy protein isolate and hydroxypropyl-α-cyclodextrin as wall materials for incorporation into sausages [161]. While the study did not directly analyze microbiological activity, the authors focused on the volatile compounds in the EO, as they are linked to its antibacterial potential. Antioxidant capacity was measured using peroxide value, acid value, and TBARS assays on the pure EO, EEO, and four EEO variations combined with commercially available antioxidants.

The results indicated that FEO sausages had poorer oxidation stability than all other samples over the storage period, showing significantly higher peroxide, acid, and TBARS values after 40, 24, and 8 days, respectively. EEO samples displayed oxidation stability comparable to those containing added antioxidants, indicating that encapsulation helped inhibit oil oxidation. The retention rate of hydroxy-α-sanshool, the compound responsible for the unique tingling and numbing sensation in Sichuan pepper oil, was measured over 56 days of storage at room temperature. The retention rate was lowest in the FEO sample (29.41 ± 4.52%) and highest in the EEO sample (62.65 ± 3.36%). EEO retention rates showed no significant differences among three of the four antioxidant-enhanced samples [161].

In situ lipid oxidation tests revealed that EEO more effectively inhibited lipid oxidation in sausages compared to FEO. Peroxide, acid, and TBARS values in EEO sausages were reduced by 37.9%, 45.2%, and 14.8%, respectively, relative to FEO sausages. Texturally, EEO incorporation improved sausage quality, reducing hardness, adhesiveness, and chewiness, while increasing springiness and cohesiveness, thereby enhancing the overall texture. Color was not significantly impacted by the inclusion of either pure or microencapsulated EO. As expected, hydroxy-α-sanshool concentrations remained significantly higher in EEO sausages than in FEO sausages after 30 days, with FEO showing a 76.56% decrease, and EEO a 68.14% decrease [161].

The articles discussed in this subsection primarily focused on EO encapsulated with polysaccharides, the most common wall material used in food applications. However, a study by Cui et al. [162] explored the effects of liposome-encapsulated clove EO in tofu, using soy lecithin (a phospholipid); cholesterol (a lipid); and polyvinylpyrrolidone (PVP), a synthetic polymer, as the encapsulating materials. This encapsulation technique enabled a controlled release of the core ingredient, triggered by the interaction between the liposomes and pore-forming toxins (PFTs), as confirmed by the authors.

Initial in vitro antimicrobial testing demonstrated the potent antibacterial effects of clove EO, achieving a 99.998% reduction in *E. coli* and a 99.999% reduction in *S. aureus* within 4 h. After 8 h, these reductions reached almost 99.999% and 99.9999%, respectively. To test the hypothesis of controlled release via PFT activation, *E. coli* (which does not secrete PFT) was used as a control, as it should not trigger the release of clove oil from the liposomes. The hypothesis was confirmed by the bacterial population results: after 96 h of incubation, the *S. aureus* population had been reduced by nearly 99.9999%, while the *E. coli* population remained unchanged, indicating no release of the EO. The gradual reduction in *S. aureus* during the incubation period supports the idea of a slow, controlled release of the encapsulated oil triggered by PFT secreted by the bacteria [162].

The capsules were also tested in situ using tofu as the food matrix. In this context, a 99.87% reduction in *S. aureus* was observed after 24 h, increasing to 99.99% after 120 h. The authors concluded that liposome-encapsulated clove EO effectively inhibited *S. aureus* in tofu, thereby extending the product’s shelf life [162].

A summary of the results from the articles reviewed in this subsection, particularly focusing on microbiological analyses, can be found in Table 3.

### 5.2. Flavoring

Using encapsulated aromatic compounds offers several advantages over using pure ingredients, particularly in terms of preserving and stabilizing the encapsulated flavoring agents after incorporated into food products. This process safeguards flavors in three key aspects: flavor retention during production and storage, flavor retention during cooking, and flavor release during eating [98]. When formulating beverages or food items intended for dissolution in liquids, ORs are often preferred due to their greater solubility in water compared to EO. However, ORs are sensitive to light, oxygen, and moisture, making it challenging to preserve their natural properties in their raw form. Unlike microcapsules designed for preservation, the encapsulated materials in flavoring capsules are generally intended for rapid release during preparation or consumption [163].

Once added to foods and packaged, volatile aromatic compounds gradually evaporate from the food matrix into the atmosphere within the package. From there, they may migrate to the outside of the package due to sorption or permeation through the material, or released when the package is opened by the consumer [98].

Moisture is a significant challenge in the flavoring industry, as it accelerates the degradation of flavor compounds. Encapsulation serves as an effective protective measure, shielding these compounds from air moisture exposure. Hydrophobic encapsulating agents, in particular, offer superior protection against moisture, enhancing the stability and longevity of the flavors. In contrast, microencapsulates with hydrophilic wall materials—designed specifically to dissolve in water—tend to break down more easily when exposed to air moisture [98].

Krishnan et al. [164] encapsulated cardamom OR using three different wall materials: gum arabic, maltodextrin, and modified starch, which were all spray-dried. The study evaluated the flavor retention, content, and stability of the powders by analyzing the volatile and non-volatile compounds in the pure OR and comparing them to the results from the microcapsules over a period of six weeks. The authors concluded that gum arabic was particularly effective in protecting the constituents of cardamom due to its film-forming capabilities and plasticity. Additionally, they noted that the free-flowing nature of the microencapsulated powders facilitated their use and incorporation within the food industry.

In baked products, microencapsulation offers several advantages, such as protecting the core ingredients from the food matrix and environmental factors, thereby reducing oxidation and thermal degradation. It also allows for the controlled release of inner compounds into the food matrix, masks undesirable flavors from bioactive agents, separates reactive ingredients until they are needed, and ensures uniform dispersion of small quantities of core ingredients by bulking them with wall materials [165]. Microencapsulated flavors are already widely used by chewing gum manufacturers, as incorporating liquid flavors can disrupt the structure of the gum matrix and compromise product quality. These flavors are typically encapsulated through coacervation, and to achieve an instant burst of flavor when chewing, the wall material must be water-soluble, releasing the flavoring agent upon contact with saliva [166]. To prolong the flavor release during chewing and enhance the consumer experience, this water-soluble matrix can be coated with a water-insoluble material, allowing for a slow release of the core ingredient as the capsules rupture during mastication and the inner wall dissolves [167,168].

A study conducted by Yeo et al. [169] demonstrated that microencapsulating flavoring baking oil through complex coacervation with gelatine and gum arabic can enhance the taste and aroma quality of frozen baked products. The encapsulated oil is engineered to release its components at higher baking temperatures while remaining intact during freezing. Gelatine facilitates the retention of the oil within the capsules at lower temperatures due to its temperature-dependent solubility. The study examined the stability of the capsules in NaCl and sucrose solutions, revealing that oil release was greater in NaCl solutions, whereas stability was preserved in sucrose solutions. The authors suggest that the integrity of the capsules in highly salted frozen foods is not a significant concern, as the salt diffusion process occurs more slowly at freezing temperatures.

The impact of various wall materials on flavor retention and stability of spray-dried volatile aroma compounds was investigated by Charve and Reineccius [170]. Their analysis showed that higher retention of individual aroma compounds occurs with increased solid content, resulting in better flavor preservation of the encapsulated compounds. At solid concentrations of 35–40%, gum arabic, modified starch, and whey protein isolate exhibited superior flavor retention due to their low viscosity, which allows for effective spray drying at these concentrations. Conversely, at lower solid concentrations (10%), soy protein isolate, sodium caseinate, and modified starch proved more effective at retaining flavor.

The study also analyzed flavor retention during storage, indicating that a higher percentage of solids leads to reduced losses of volatiles over time. Gum arabic provided the best retention for three out of the four aroma compounds studied, particularly aldehydes. In contrast, limonene, a volatile monoterpene, was better retained by soy protein isolate, whey protein isolate, and sodium caseinate—proteins that produce capsules less permeable to oxygen, thereby limiting limonene oxidation and enhancing flavor retention [170].

Finally, the authors observed color differences related to browning during storage, noting that modified starch and gum arabic did not undergo browning, while whey protein isolate exhibited the most browning, followed by sodium caseinate and soy protein isolate. Interestingly, solids concentration did not influence the browning of the powders. In conclusion, careful consideration of the oil composition intended for encapsulation is essential to achieve optimal aroma retention [170].

An optimization study was conducted to develop spray-dried microcapsules using maltodextrin and Persian gum for the encapsulation of cinnamon EO, aimed at achieving quick-release mechanisms upon contact with saliva [171]. Various ratios of the two polymers were analyzed, revealing that the optimal combination was 8.26% maltodextrin and 1.74% Persian gum. This formulation demonstrated the highest values for powder recovery, EO release, and encapsulation efficiency. The encapsulation effectively retained the oil’s major components, which were analyzed using gas chromatography with mass spectrometry (GC-MS). Unlike pure EO, which is hydrophobic, the optimized formulation exhibited high solubility in water (82.3%), a crucial attribute for applications in the food industry. An in vitro oil release analysis confirmed the rapid release of the desired component, with nearly 70% of the core material being released within 60 s. Furthermore, the antioxidant activity of the encapsulated EO was comparable to that of the pure EO, indicating that the formulation did not significantly diminish the oil’s scavenging capacity.

Fadel et al. [172] evaluated the effects of encapsulated cinnamon EO on the aroma quality and stability of biscuits. The EO was extracted from cinnamon bark and used to formulate two flavoring products: a liquid flavor comprising propylene glycol and 30% *v*/*v* cinnamon EO, and an encapsulated flavor using maltodextrin as a wall material at a concentration of 3% *w*/*w*. Three biscuit batches were created: a control without flavoring, one with the liquid flavor, and another with the encapsulated flavor, which replaced 10 g of sugar in the recipe. The retention of cinnamaldehyde was higher in the biscuits with encapsulated oil (95.65%) compared to those with the liquid flavor (87.11%). This trend was also observed for eugenol and β-caryophyllene. Although the concentrations of these three compounds decreased during storage in both samples, the encapsulated biscuits maintained higher levels with a slower decline.

The authors analyzed the stability of 57 volatile compounds during baking and over a 90-day storage period for the three biscuit samples. The encapsulated sample exhibited the highest levels of Maillard reaction products (71.79%), followed by the liquid flavor sample (52.63%) and the control (51.54%). Additionally, lipid oxidation was lower in the samples containing cinnamon oil, with those incorporating encapsulated oil demonstrating better oxidation prevention than those with non-encapsulated oil. The total yield of lipid degradation products was lower in biscuits made with encapsulated oil compared to the control and those containing non-encapsulated oil after 90 days of storage [172].

A sensory analysis of the two formulations with added cinnamon EO was conducted at different storage intervals. The color did not show significant differences over time, but the aroma and taste quality were superior in the encapsulated sample. Both biscuit types exhibited similar crispiness, although the encapsulated oil sample received slightly higher scores throughout the evaluation period. Overall acceptability was higher in the biscuits made with EEO, leading to the conclusion that EEO is a promising additive for cookie doughs, serving not only as a flavoring agent but also as an antioxidant [172].

Kausadikara et al. [44] optimized the formulation of microencapsulated lemon EO by investigating gum arabic, maltodextrin, modified starch, and their binary and ternary blends as wall materials, using the spray-drying technique. The resulting atomized powders were intended for use in an instant iced tea premix. The authors also incorporated two additional EOs, ginger and cardamom, into the formulation development. To establish the optimal process conditions for spray drying, a preliminary assessment was conducted to evaluate the effects of oil loading and atomization temperature.

The effective encapsulation efficiency (calculated based on the theoretical loading of oil) was determined for lemon oil microcapsules using pure gum arabic as the wall material at various oil concentrations and inlet temperatures (150, 160, 175, and 180 °C). The optimal combination was found to be 10% oil content at an air inlet temperature of 175 °C. The effective encapsulation efficiency increased with higher temperatures for capsules containing 20% and 30% oil concentrations. Thirteen formulations with different compositions and ratios were developed and analyzed based on various parameters [44].

The authors used the conditions that gave optimum encapsulation efficiency to investigate the effects of wall-material formulations, which revealed that higher concentrations of gum arabic led to increased encapsulation efficiencies, while maltodextrin had a negative impact on this parameter. In binary blends, the optimal combination was a 50:50 ratio of gum arabic and modified starch, achieving an effective encapsulation efficiency of 83.60%. The best ternary blend comprises equal parts of each encapsulating material, yielding an effective encapsulation efficiency of 77.79%. All samples demonstrated high encapsulation efficiency, exceeding 93%. The highest efficiency was observed in the 75:25 gum arabic and maltodextrin blend (98.95%), followed by the 25:75 gum arabic and maltodextrin blend (98.69%) and the ternary blend of one-sixth gum arabic, four-sixths maltodextrin, and one-sixth modified starch (98.43%). The lowest encapsulation efficiency was recorded for the 75:25 gum arabic and maltodextrin sample, which had an effective encapsulation efficiency of 93.10% [44].

An unknown formulation of the encapsulated lemon oil was incorporated into an iced tea drink at a 1% level. This was divided into three temperature categories (4, 28, and 45 °C) and tested at three different intervals over a three-week period for stability analysis. Samples were evaluated by an expert panel for sensory analysis. No significant changes in the appearance of the drinks were noted over the three weeks, although the capsules introduced slight turbidity and sediment during storage. The capsules exhibited controlled release of lemon flavor in the drink, maintaining flavor intensity effectively. The iced tea with incorporated microencapsulated lemon oil preserved its sensory characteristics for three weeks at 45 °C, which is equivalent to one year of stability at room temperature. Finally, the concentration of lemon oil powder in the drink mix was reviewed, and it was concluded that the formulation with the highest score on a hedonic scale, assessed by seven expert panel members, contained 1.5% lemon powder [44].

Hernández-Fernández et al. [53] developed vanilla OR microcapsules through complex coacervation followed by spray drying, aiming to protect volatile compounds for later incorporation as flavoring agents in food products. The OR was extracted from cured vanilla beans using supercritical CO_2_ as the solvent. The optimal coacervate formulation was achieved with 0.34% chitosan and 1.7% gum arabic, at a ratio of OR to wall material of 1:2.5. Subsequently, the coacervates were spray-dried at an inlet temperature of 100 °C.

Physicochemical characterization revealed satisfactory water activity levels (0.13), moisture content (3.80%), retention efficiency (84.89%), and encapsulation efficiency (69.20%). Scanning Electron Microscopy (SEM) images indicated that most microcapsules were round and smooth-surfaced, with only a small percentage exhibiting fractures, suggesting effective encapsulation of the OR. A comparison of the microcapsules before and after spray drying showed no alteration or degradation of the vanilla compounds. Therefore, it can be concluded that vanilla OR can be successfully microencapsulated using complex coacervation and spray-drying techniques for application as flavoring in food matrices [53]. Combining these two encapsulation methods allows for precise control over microcapsule size and ensures controlled release in high-water-content matrices, all while maintaining relatively low operational costs [12,133].

As evidenced by the findings of the articles presented in this section, microencapsulated EOs and ORs used as flavorings offer the added advantage of preserving food products and extending their shelf-life. The constituents responsible for the biological activity of these extracts, namely terpenes, flavonoids, phenols, are also responsible for the aroma, owing to their small structure and volatility [173,174]. Artificial flavorings consist of synthetized molecules designed to recreate the ones found in nature and that provide specific flavors [175]. However, they lack the complexity of constituents and synergetic effects between bioactive compounds that gives EOs and ORs their potent antioxidant and antimicrobial properties [176].

From the perspective of salt reduction, microencapsulated ORs present a promising alternative to sodium chloride and other salt substitutes. A recent 2024 study indicated that the incorporation of three microencapsulated ORs into sauces could reduce salt content by 25% to 50% with minimal impact on physicochemical and organoleptic characteristics [177]. These ORs, formulated with a blend of aromatic plants and spices and microencapsulated with inulin, were added to mayonnaise, mustard, and ketchup with reduced salt content. A mineral analysis revealed significant differences between the control sauces and those containing microcapsules. A 50% reduction in salt was achieved in most sauces, while two mayonnaise samples showed a 25% reduction due to lower initial salt levels.

Viscosity measurements indicated no significant differences in the mustard samples, whereas the mayonnaise samples with microencapsulated OR exhibited higher viscosity values compared to the control. Ketchup consistency varied significantly among samples, with the control exhibiting the highest viscosity. Color differences between the sauces and their respective controls were less pronounced in the mayonnaise, followed by mustard, and most noticeable in ketchup. The authors attributed this color effect to the light yellow hue of the microcapsules, which impacted darker sauces more noticeably [177].

Sensory analysis conducted by a panel of untrained consumers revealed that the formulations of mayonnaise and ketchup containing ORs closely resembled the control, with no significant differences detected. However, the mustard sample with microcapsules showed a significant reduction in flavor and salt perception compared to the control, resulting in a lower overall score. In conclusion, encapsulated ORs from spices and plants hold promise as a salt substitute for flavoring purposes, although further research is needed to explore the impact of various spice and plant mixtures on salt perception [177].

A 2021 article [178] evaluated the effects of sodium chloride reduction by creating three different sausage samples: one that replaced 50% of sodium chloride with microencapsulated OR, another that substituted 50% of sodium with potassium chloride (KCl), and a third that eliminated 50% of the salt entirely. A sensory analysis of all four sausage samples, including the control (100% NaCl), indicated that salty flavor was the only taste descriptor significantly affected by the sodium reduction. A texture analysis also revealed notable differences, with the reduced-sodium sausages being less firm. However, this difference in firmness disappeared after five weeks of storage. Additionally, sausages made with microencapsulated OR exhibited reduced lipid oxidation throughout the storage period.

Another study focused on adding microencapsulated OR (MO) to smoked rainbow trout as a means to reduce sodium content [179]. This study included one control sample (100% NaCl) and five reduced-salt samples: 75:25 NaCl:MO, 50:50 NaCl:MO, 50:50 NaCl:KCl, 50:50 NaCl:KCl with a bitterness masking agent, and 25:75 NaCl:KCl with a bitterness masking agent. Unlike the previous study on fish sausages, no significant differences in texture were observed among the trout samples. However, a sensory analysis indicated that the 50:50 NaCl:MO sample was perceived as significantly saltier than all other samples, including the control with 100% NaCl, while exhibiting notably higher bitterness values. Consequently, the formulations containing 25% microencapsulated ORs and the two samples with a 50:50 NaCl:KCl ratio were deemed effective solutions for reducing sodium in smoked trout, as their sensory attributes were more balanced.

In conclusion, the potential of microencapsulated ORs as salt replacers is promising; however, further research is needed to optimize formulations and ratios for a more balanced product. Despite the growing body of research on microencapsulated EOs and ORs, studies exploring their direct applications in food products or beverages remain limited. This is particularly evident when compared to research focused on the microencapsulation of these extracts and their individual characteristics, or the application of pure EOs and ORs in food without encapsulation. While encapsulating flavors may sometimes diminish their intensity and taste perception, the protective benefits afforded to bioactive compounds—along with their controlled release during consumption—make microencapsulation a valuable approach for incorporating natural additives into food products. This is especially appealing to consumers seeking enhanced health benefits without resorting to supplements or pills.

### 5.3. Coloring

The spices that are most commonly used as natural colorants in food include paprika, red pepper, mustard, parsley, ginger, and turmeric [180]. Molecular compounds such as carotenoids, flavonoids, betalains, chlorophylls, and curcumin are responsible for the colors found in nature and are of particular interest to the food industry [181,182]. Carotenoids are a class of pigments that fall into two categories: carotenes (α- and β-carotene and lycopene) and xanthophylls (lutein, zeaxanthin, astaxanthin, fucoxanthin, and cryptoxanthin). These pigments provide yellow, orange, and red colors [183]. Flavonoids can be divided into anthocyanins, flavonols, and chalcones and can provide a wide range of colors, such as white-cream, orange, red, blue, and purple [182,184]. Betalains are known for their red, yellow, and orange hues and are present in amaranth (*Amaranathus tricolor)* and beets (*Beta vulgarius*) [185]. Chlorophyll, whose molecular structure provides several shades of green, can be extracted from alfalfa (*Medicago sativa*) and is present in almost every plant and seaweed [182,186,187]. Curcumin is a phenolic component mainly found on turmeric (*Curcuma longa*), which has been used for centuries for its strong yellow color [181,188]. Unlike EOs, which are colorless volatile extracts, ORs contain a non-volatile resin fraction, from which the pigments responsible for their characteristic colors are derived [114]. However, the pigments in ORs are sensitive to high temperatures and oxygen, leading to degradation during processing and storage. Encapsulation can protect these pigments, enhancing their stability before and after application in food products [189].

A search of several academic online platforms was conducted to find research articles focused on optimizing microencapsulated ORs for use as food colorants, specifically those tested directly in food products or beverages. Unfortunately, only one relevant paper was identified. A more thorough investigation revealed an article that encapsulated an astaxanthin extract for use in a beverage model. Although the authors referred to this extract as a lipid extract rather than an OR, the extraction methods and solvents used are similar to those employed in OR extraction. Therefore, lipid extracts obtained with food-grade solvents have been included in this section to expand the pool of relevant literature [190,191]. However, studies examining the stability of coloring compounds in microencapsulated ORs in vitro are more readily available. Research into the application of microencapsulated ORs in food matrices remains underdeveloped, suggesting an opportunity for further exploration to develop tailored formulations. Such formulations could be designed for specific chemical environments, ensuring their dispersibility in the medium while providing strong, stable colors without compromising other organoleptic properties that might affect consumer acceptance [192].

Kshirsagar et al. [193] evaluated the color capacity of turmeric OR, both in its pure form and when microencapsulated, in sorghum flour extrudates. In this 2010 study, the authors compared the differences between pure OR and microencapsulated OR in colored products prior to extrusion. An empirical statistical modelling technique was initially employed to optimize the moisture content, die temperature, and screw speed parameters to maximize color values. The experimental phase involved preparing three distinct samples: one incorporating pure turmeric OR, another using turmeric OR microencapsulated with gum arabic, and a third microencapsulated with n-OSA starch. For microencapsulation, emulsions were prepared with 30% (*w*/*w*) of the mentioned wall materials, 10% of the weight of the wall materials of OR, and 1% of pullulan relative to the total solution. One percent of the microcapsules was added directly to the flour, mixed, and then extruded. In contrast, 1% of the pure OR was first dissolved in propylene glycol, mixed into the flour, and subsequently extruded. The color of the extrudates was evaluated based on the interaction of three independent variables: moisture content, die temperature, and screw speed, comparing predicted values with empirical observations. The analysis concluded that microencapsulated ORs exhibited superior color values and stability under the conditions experienced during the extrusion process.

Astaxanthin, a carotenoid found in the shells of shrimp, was quantified and evaluated by Bassijeh et al. [190] for its potential as a natural colorant in beverages. A lipid extract was obtained from frozen shrimp and microencapsulated through freeze-drying, using whey protein isolate (WPI) and Persian gum as encapsulating agents. Various ratios of wall materials were tested (WPI ratios of 1:2, 1:3, and 1:4). All samples exhibited high solubility in water, ranging from 94.32 to 95.59 g/100 g, with encapsulation efficiency being highest in the 1:4 ratio (49.85 ± 0.37 g/100 g). The lyophilized capsules displayed a light red color due to the presence of astaxanthin. The L* values, which measure lightness, decreased as the concentration of Persian gum increased, attributed to its darker hue. The parameters a* and b* remained consistent across the different samples.

Table 4 presents a selection of published articles in which ORs from various sources (both plant- and animal-based) were microencapsulated and their coloring properties examined. The formulations tested involved different wall materials and ratios, as well as varying inlet temperatures for some of the spray-dried capsules. The studies evaluated color parameters, concentration, stability, and oxidation of the compounds responsible for the vibrant colors of the corresponding OR, along with their stability under temperature, light, pH, and storage conditions. Water solubility was noted by several authors, as this parameter is crucial in the food industry for facilitating the incorporation and distribution of the powder in the desired food or beverage product. Additionally, emulsion stability is a key attribute for predicting the visual stability of the product under various storage conditions, particularly in the beverage sector [194].

### 5.4. Functional Foods

Nutraceuticals and functional foods have gained significant popularity in the food industry, particularly within the healthy food market, appealing to consumers seeking products with added health benefits. While these terms are often used interchangeably, they are distinct. Functional foods are defined as products that resemble traditional foods or beverages but offer additional health advantages. In contrast, nutraceuticals are derived from bioactive substances extracted from food and are typically delivered in the form of pills, capsules, or liquids as dietary supplements [57,201]. Both categories are commonly associated with claims of reducing the risk of various health issues, including cancer, heart disease, obesity, digestive problems, diabetes, insomnia, and headaches [202,203]. A wide range of bioactive compounds found in EOs and ORs have been identified and studied, with these antioxidant agents demonstrating numerous health benefits, such as anticancer, anti-inflammatory, antimutagenic, and cardioprotective properties [204].

When incorporated into food products, these plant extracts can support nutrition and health claims, as outlined by current regulations. In the EU, Regulations No. 1924/2006 [205] and No. 432/2012 [206] establish permitted claims, while in the US, the 1990 Nutrition Labelling and Education Act (NLEA) [207]; the 1997 Food and Drug Administration Modernization Act (FDAMA) [208]; and the FDA’s guidance document titled “Guidance for Industry: Interim Procedures for Qualified Health Claims in the Labelling of Conventional Human Food and Human Dietary Supplements” [209] dictate the health claims that can be made on food product and dietary supplement labels [210].

The microencapsulation of EOs and ORs provides protection within the food matrix and facilitates the effective delivery of bioactive compounds to the body during consumption, ensuring their stability throughout processing and storage. As discussed in previous sections, encapsulation enhances solubility in hydrophilic matrices and shields reactive molecules from other compounds present in the incorporated product. These molecules often exhibit low bioavailability due to poor solubility and the potential for enzymatic degradation within the body upon ingestion. Therefore, protecting these compounds is crucial for ensuring their targeted release [66].

The development of functional foods and nutraceuticals derived from natural sources of bioactive substances is on the rise, driven by the rapidly growing market catering to an aging population in developed countries and health-conscious consumers [211]. Numerous articles examine the health benefits of EOs and ORs, yet there are few studies specifically focused on the effects of microencapsulation of these extracts in food and beverages, as well as their stability, release mechanisms, and overall impact on the body.

In a 2015 study by Li et al. [212], lycopene-rich tomato OR was encapsulated to protect its disease-fighting properties from degradation during digestion. The OR was encapsulated by mixing with soy protein isolate and gum arabic conjugates, followed by spray drying at 140 °C. The goal was to prevent the capsules from releasing lycopene in the stomach, allowing it to reach the intestine for optimal absorption. The authors conducted controlled release analyses under acidic conditions (pH 1.2) to simulate gastric fluids, followed by testing in a basic solution (pH 7.4) to mimic intestinal fluids. The results demonstrated that the wall materials effectively protected the core in acidic conditions, releasing 93% of lycopene under simulated intestinal conditions. Additionally, encapsulation increased lycopene retention by 33% and 80% relative humidity, even in the presence of light, compared to non-encapsulated OR.

Tomato OR has demonstrated significant nutraceutical potential due to its high lycopene content, which contributes to its antioxidant and antimutagenic properties, as highlighted in a 2009 study [213]. This OR was derived from non-commercial red tomatoes, making it a valuable resource for the food industry by enabling profit generation from previously unmarketable vegetables.

The application of a versatile medicinal plant in the food industry was also explored through the use of spray-dried microencapsulated myrtle EO [214]. Known for its traditional use in treating gastric ulcers, myrtle EO capsules were evaluated for their gastroprotective effects against gastric lesions. In animal studies, maltodextrin-encapsulated myrtle EO was shown to reduce lesion formation in the gastric mucosa without causing toxicity at the administered concentrations. Thus, microencapsulated myrtle EO could serve as a functional food ingredient for individuals suffering from ulcers or conditions related to excessive stomach acid.

In a more recent study, ultrasonic emulsified microencapsulated lemon balm EO was incorporated into flavored yogurt to create a functional food product [215]. Yogurt, already recognized for its health benefits, can be further enhanced with additional therapeutic properties, helping it stand out in a competitive market. Microcapsules made from whey protein isolate and sodium caseinate exhibited a good release rate of EO throughout storage in the yogurt. However, sensory analysis indicated that panelists were unable to distinguish between the control yogurt and the yogurt enriched with microcapsules, suggesting that the flavor of the EO was not perceptible.

A direct application of microencapsulated garlic EO was tested in an acidic beverage model to create a drink that retains the health-promoting properties of garlic without its characteristic taste [216]. Garlic is known for its diverse array of bioactive compounds, including allicin, which the authors used to monitor the sulfur content in the powders. The β-cyclodextrin and pectin capsules demonstrated superior solubility in water compared to pure garlic oil, which is advantageous for their incorporation into beverages.

The release of garlic EO from the microcapsules was evaluated in the acidic beverage model, revealing that the release rate increased over time, rising from 13.4% after 30 days to 23.3% after 60 days. The release was also assessed under simulated gastric and intestinal conditions. The wall material effectively inhibited release in the stomach acid model, with a maximum release of approximately 3%. In contrast, the release rate in intestinal conditions was nearly instantaneous, with almost 100% of the content released within 60 min. This indicates that these microcapsules can be successfully integrated into functional beverages, allowing garlic’s bioactive compounds to survive gastric conditions and be absorbed in the gastrointestinal tract [216].

Additionally, three beverage samples were tested and their sensory attributes compared: a commercial orange juice without EO (control), an orange juice with pure EO added, and an orange juice with microencapsulated EO. Ten semi-trained panelists evaluated the samples based on smell, taste, turbidity, and ease of swallowing at three storage intervals (0, 30, and 60 days). Initially, the drink with pure garlic oil received the lowest scores for taste and aroma, while the drink with microencapsulated oil scored similarly to the control. Over time, the taste, smell, and turbidity scores of the encapsulated sample decreased, ultimately scoring lower than the control. However, no significant differences were observed in terms of ease of swallowing among the samples throughout the storage period. In conclusion, microencapsulated garlic EO presents a promising application in the functional drinks market [216].

## 6. Flavor Masking for Specific/Particular Applications

Encapsulation effectively reduces and masks the strong taste of plant extracts that may be undesirable in certain food products. If the primary goal of incorporating microencapsulated EO in a recipe is to enhance the stability and shelf life of the final product, then reducing the flavor intensity can serve as a viable alternative to synthetic preservatives without significantly compromising consumer acceptability [135].

A 2018 study on thyme EO in romaine lettuce demonstrated that microencapsulated EO yielded better overall results compared to pure oil [158]. The effects of β-cyclodextrin-encapsulated thyme EO were compared with pure EO and untreated lettuce leaves in terms of weight loss, color, total phenolic content, total flavonoid content, antioxidant activity, microbiological quality, and organoleptic quality. The microencapsulated oil samples outperformed the pure oil lettuce in every chemical and biological analysis, showing increased levels of flavonoids, phenolic compounds, and antioxidant activity, whereas the pure oil led to a decrease in these values. In terms of flavor masking, the lettuce treated with pure EO was negatively affected by the strong aroma of the oil’s volatiles, a perception that was largely absent in the samples containing encapsulated oil—except for those with a higher oil concentration. Overall, the lettuce sample with the lowest percentage of encapsulated EO was the only one to receive an acceptable visual score at the conclusion of the 12-day analysis.

In a previously mentioned study [177], microencapsulated OR successfully reduced salt content in mayonnaise and ketchup without significantly altering the aroma or taste compared to the control sauces. Additionally, another article explored cinnamon extract microcapsules obtained through complex coacervation and tested their effects in ice cream formulations [217]. Two samples were prepared: one with gelatine and gum arabic, and another with gelatine and κ-carrageenan. Sensory analyses were conducted on three ice cream samples containing cinnamon extract: a control with non-encapsulated cinnamon extract and the two samples with different wall-material formulations. The results confirmed the flavor-masking capabilities of encapsulation, with all panelists indicating that the ice cream with non-encapsulated extract had the strongest flavor and was more astringent.

## 7. Future Perspectives and Conclusions

EOs and ORs are valuable sources of volatile and non-volatile compounds that can be incorporated into food products and beverages as natural additives. However, these compounds are sensitive to light, oxygen, moisture, and temperature, making them prone to oxidation. Encapsulation offers an effective method for protecting these essential molecules within food matrices and packaging.

Various microencapsulation techniques have been studied and implemented in the food industry, but research focusing on the incorporation and effects of microencapsulated EOs and ORs in specific food products and beverages remains limited. While newly developed microencapsulation technologies could provide solutions for creating more stable capsules, the high costs associated with these techniques pose a challenge for the industry. Nonetheless, these advancements could be particularly beneficial in the development of functional foods, where consumers may be willing to justify a price increase for enhanced stability and health benefits.

There is still significant scope for research into different wall-material formulations for encapsulating EOs and ORs for use in food and beverages. As common wall materials become more expensive, there is an industry push to find alternatives with similar properties at lower costs. To reduce the cost of EO and OR microcapsules, research could focus on the development of novel encapsulating agents with enhanced encapsulation efficiency and improved protective properties, or the optimization of formulations using currently available wall materials in the industry. Potential advancements could be achieved by investigating key factors, such as the optimal ratios of encapsulating agents with complementary properties to address individual weaknesses; tailoring formulations to specific EOs or ORs, considering the diverse compositions of their volatile and bioactive molecules; and designing formulations optimized for the intended application in food products. This includes considerations for use in liquid versus solid matrices; low-moisture versus high-moisture products; and specific release profiles—whether immediate, sustained, or a combination—depending on their functional role in the final product.

Despite several microencapsulation techniques being well-established in the industry, opportunities remain for improving encapsulation parameters and existing equipment. Simple and cost-effective upgrades, such as incorporating an air dehumidifier unit into a spray dryer, can reduce liquid feed drying time and potentially lower the drying air temperature, thereby preserving sensitive compounds. Critical parameters, including inlet air temperature, air flow rate, feed flow rate, and the type of atomizer, significantly influence the quality of the produced spray-dried powder. These factors should be systematically investigated and optimized for each specific product to achieve the desired encapsulation efficiency and powder characteristics. Other scalable techniques with favorable encapsulation properties but lower production yields could also contribute to cost reduction of the microcapsules. To achieve this, research should focus on minimizing product losses and accelerating the capsule production process while maintaining high-quality standards.

Beyond flavor enhancement, these extracts can also help reduce sodium content in food—an area of great relevance to the industry. The extracts discussed in this review demonstrate substantial potential across various sectors of the food industry, serving as natural preservatives, colorants, flavorings, and sources of bioactive molecules with health benefits. The growing consumer concern over synthetic additives and the search for natural alternatives can encourage both industry stakeholders and investors to explore the use of EOs and ORs as food additives.

## Figures and Tables

**Figure 1 foods-13-03873-f001:**
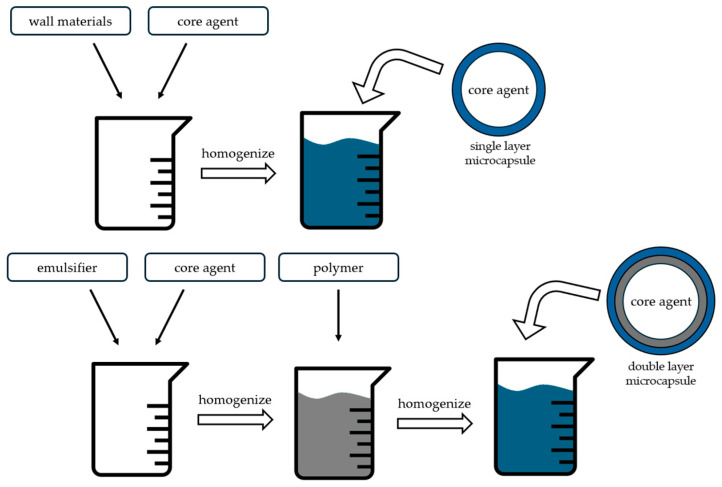
Example of a methodology to produce single- and double-layer microcapsules.

**Figure 2 foods-13-03873-f002:**
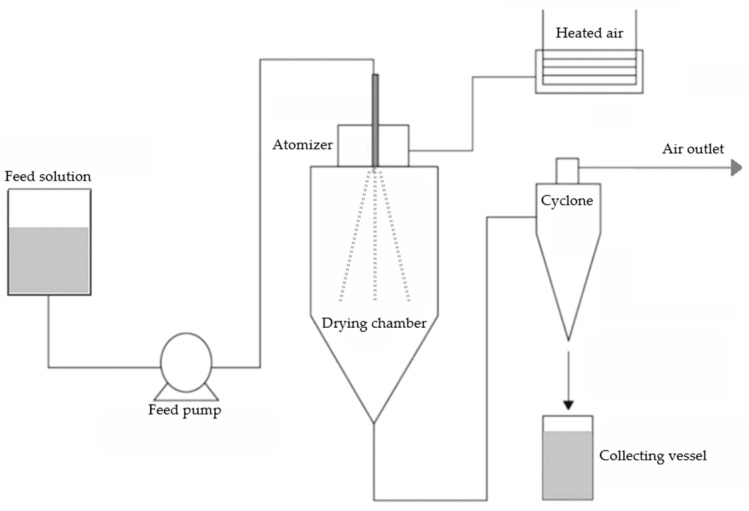
Schematic illustration of a spray dryer, adapted and modified from [61].

**Figure 3 foods-13-03873-f003:**
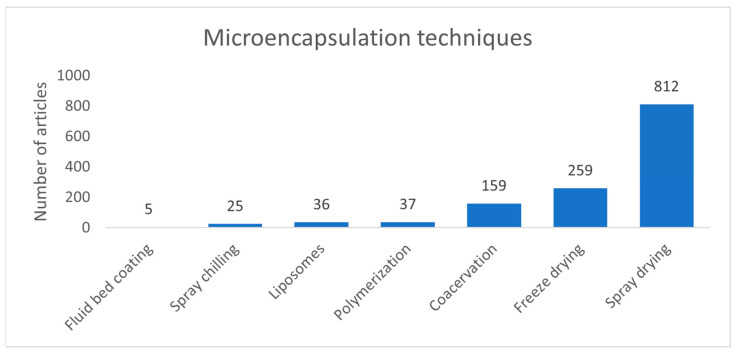
Number of articles addressing distinct microencapsulation techniques since 2010 on Scopus database.

**Table 1 foods-13-03873-t001:** Techniques for microencapsulation of EOs and ORs in food applications.

Nature	Technique	Core Material	Wall Material	Function	Reference
Physical	Spray drying	Sichuan pepper OR	Sodium octenyl succinate starch, tea polyphenols	Flavoring	[42]
		Paprika OR	Gum arabic, modified corn starch, and pregelatinized waxy corn	Colorant	[43]
		Lemon, ginger, and cardamom EOs	Gum arabic, maltodextrin, and modified starch	Flavoring	[44]
	Freeze drying	Thyme EO	Whey protein concentrate and sodium alginate	Preservation	[45]
		African basil EO	Gum arabic and cassava starch	Preservative	[46]
	Spray chilling	Cinnamon and paprika ORs	Palm oil	Preservative	[47]
	Extrusion	Rosemary EO	Alginic acid and calcium chloride	Preservative	[48]
	Fluidized bed coating	Dill EO	Maltodextrin, gum arabic, and microcrystalline cellulose	Preservative	[49]
	Supercritical fluids	Rosemary EO	Pluronic^®^ F-127 (BASF, Ludwigshafen, Germany) and Pluronic^®^ F-88 (BASF, Ludwigshafen, Germany)	Preservative	[50]
Chemical	Interfacial polymerization	Osmanthus EO	Amisoft® GCS 11 (Ajinomoto, Tokyo, Japan), Desmodur^®^ N 100 (Bayer, Leverkusen, Germany)	Flavoring	[51]
	In situ polymerization	Thyme EO	SLS, T80, P127, PVA	Insect repellent (packaging)	[52]
Physicochemical	Coacervation	Vanilla OR	Chitosan and gum arabic	Flavoring	[53]
		Tea tree EO	Gelatine, gum arabic, chitosan, sodium alginate, and soybean protein isolate	Preservative	[54]
	Ionic gelation	Marjoram EO	Sodium alginate and whey protein isolate	Preservative	[55]
	Liposomes	Curry plant EO	Soy lecithin and cholesterol	Preservative	[56]

**Table 3 foods-13-03873-t003:** Analysis of microbiological results in studies on microencapsulated EOs for food applications.

				Microbiological Analysis		
Product	Encapsulation Technique	Core Material: EO	Wall Material	In Vitro	In Situ	Reference
Cake	Precipitation, anti-solvent precipitation, and electrospraying	Orange	β-cyclodextrin, zein	FEO had AA against *Aspergillus*	FEO and EEO delayed fungal development	[154]
Cake	Co-precipitation and vacuum filtration	Orange	β-cyclodextrin	FEO had better AA against *A. Flavus* than EEO; EEO only had AA at the highest concentration tested	FEO had AA; EEO did not have AA	[155]
Cake	Complex coacervation	Thyme	Gelatine, gum arabic	EEO had better AA than FEO	EEO had better AA than FEO	[156]
Meat burger	Spray drying	Thyme	Maltodextrin, sodium casein	FEO and EEO had similar AA	EEO had better AA than FEO against *E. coli*; FEO did not have AA against *E. coli;* FEO had better AA than EEO against thermotolerant coliforms	[157]
Lettuce	Co-precipitation	Thyme	β-cyclodextrin		EEO had better AA than FEO	[158]
Fresh dough	Spray drying	Rosemary	Modified starch, maltodextrin	FEO had AA with every tested concentration	EEO had better AA than FEO	[159]
Minced meat	Spray drying	Garlic	Maltodextrin, gum arabic		Highest EEO concentration had satisfactory AA	[160]
Sausages	Freeze drying	Sichuan Pepper	Soy protein isolate, hydroxypropyl-α-cyclodextrin			[161]
Tofu	Liposomes	Clove	Soy lecithin, cholesterol, PVP	FEO had AA against *E. coli* and *S. aureus*; EEO had AA against *S. aureus* but not *E. coli*; Confirmation of controlled release triggered by PFT	EEO had AA against *S. aureus*	[162]

AA, antimicrobial activity; FEO, free essential oil; EEO, encapsulated essential oil; PFT, pore-forming toxins.

**Table 4 foods-13-03873-t004:** Studies evaluating pigments and color parameters in microencapsulated ORs.

Core Material: OR	Encapsulation Technology	Wall Material	Ratios/Temperature	Conclusions	Reference
Astaxanthin	Spray drying	GA, WP, MD, IN	100% GA, 100% WP, 50:50 GA:IN, 50:50 GA:MD, 50:50 GA:WP, 25:75 GA:IN, 25:75 GA:MD, 25:75 GA:WP	The WP sample dissolved in water exhibited a stronger, deeper orange color than MD capsules; WP sample had higher astaxanthin stability and AA retention. WP samples had higher color stability at different pH values. Emulsion stability measured at different pH supports its application in instant powder drinks. Authors selected 100% WP microcapsules as the best formulation.	[195]
Astaxanthin	Spray drying	GA, MD	100% MD, 100% GA, 50:50 MD:GA	MD showed higher lightness values, while GA showed lower hue values. All samples had very high EE (94 ± 1%). MD showed higher water solubility (97 ± 3%) than GA and MD:GA (92 ± 1%). EOR showed lower lipidic stability at the beginning of storage (1 month). Astaxanthin content and color remained constant during storage period (110 days). GA showed higher astaxanthin bioaccessibility (~55%), followed by MD and MD:GA (~32%) and then FOR (~5%).	[191]
Paprika	Spray drying	GA, SPI	100% GA, 100% SPI; Inlet temperatures: 160, 180 and 200 °C	Higher inlet temperatures led to higher carotenoid retentions; GA showed good stability at lower a_w_ levels (a_w_ = 0.108), while it had poor result at higher a_w_ levels (a_w_ > 0.318). At a_w_ = 0.743, GA capsules disintegrate. SPI showed better stability at higher a_w_ levels (a_w_ = 0.743). SPI capsules were stable through different a_w_ values.	[189]
Paprika and cinnamon 1:1 mixture	Spray drying	MD, WPI	100:0, 0:100; 50:50, 75:25, 25:75 MD:WPI	50:50 and 75:25 MP:WPI showed higher EE (95.19 and 96.30%). 100:0 MD:WPI showed lowest solubility (65.94%), while 75:25, 50:50 and 0:100 had highest solubility (92.17, 90.69, and 91.97%, respectively). All samples’ emulsions remained stable for 1 h, expect for 100:0. While 100:0 showed a darker, reddish color, others showed an orange color with higher luminosity. Color differences were higher at higher temperatures (a* values reduction was greater at 45 °C than 25 °C). While 100:0 showed a higher carotenoid concentration on day 0, it showed the lowest at the end of storage (90 days) at 45 °C. Carotenoid protection at 45 °C increased with the increase in WPI. At 25 °C, 75:25 MD:WPI showed better carotenoid protection. Authors found 75:25 MD:WPI formulation to be the best, with well-balanced qualities for food applications.	[196]
Turmeric	Freeze drying	G, MS	43.3:1, 22:1, 30:1, 13:1, 10:1 S:G	Curcumin remained stable at low temperatures (T = −20 °C) during 35 days of storage. At higher temperatures, there was a reduction in curcumin compounds (color loss of 14.5% at 25 °C and 22.8% at 60 °C after 35 days of storage). Curcumin loss was 3.2 times higher with the presence of light at 25 °C. Changes in the total color parameters confirmed major losses in the presence of light. Color was not affected as much at 20 °C and 25 °C. Light was the primary cause for color degradation.	[197]
Turmeric	Spray drying	GA, MD	100:0, 0:100, 50:50, 75:25, 25:75 MD:GA; Inlet temperature: 150, 175 and 200 °C	As GA concentration increases, emulsion stability indices, viscosity, EE, and curcumin content also increase. Ideal inlet temperature was 175 °C. EOR delayed curcumin degradation by light exposure, heat, and during storage, compared to FOR. EOR had controlled release behavior over a 1-week period, under basic, acid, and neutral pH values.	[198]
Rosa Mosqueta	Spray drying	S, G	100% S (T = 150 °C), 100% G (T = 100 °C)	Recovery of carotenoid pigments was higher in G sample than S sample. Higher drying temperature in S sample led to higher losses in carotenoids. S showed higher degradation rates of the major carotenoid pigments and lower half-life than G capsules.	[199]
Red chili	Spray drying	MD, WPC, MG	1/6:1/6:2/3, 2/3:1/6:1/6 WPC:MG:MD; 2:1 and 4:1 wall to core ratio	Single, binary, and ternary blends of WPC, MG, and MD parameters were mathematically analyzed to select the best blends to spray dry. Lower OR concentration increased E_a_ for oxidation. Blend with higher MD showed higher E_a_. Hence, 1/6:1/6:2/3 WPC:MG:MD with a 4:1 wall-to-core ratio was the formulation with higher stability	[200]

A_w_, water activity; E_a_, activation energy; EE, encapsulation efficiency; EOR, encapsulated oleoresin; FOR, free oleoresin; G, gelatine; GA, gum arabic; IN, inulin; MD, maltodextrin; MG, mesquite gum; MS, modified starch; S, starch; SPI, soy protein isolate; WPC, whey protein concentrate; WPI, whey protein isolate.

## Data Availability

No new data were created or analyzed in this study. Data sharing is not applicable to this article.

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
