# Peer review of "Microencapsulation of Essential Oils and Oleoresins: Applications in Food Products"

_foods, 2024, doi:10.3390/foods13233873_

Round 1
Reviewer 1 Report
Comments and Suggestions for Authors
Os autores apenas descrevem os objetivos da revisão no resumo. Os objetivos devem ser citados no final da introdução.
Para tornar a leitura mais atrativa e compreensível, é importante que os autores incluam figuras ilustrativas. Por exemplo, uma figura mostrando que as paredes das microcápsulas podem ser de uma ou várias camadas.
Na Tabela 1, outras técnicas importantes que não foram citadas poderiam ser incluídas. Por exemplo, nas técnicas físicas, como liofilização e fluido supercrítico.
Além disso, como a técnica de secagem por pulverização foi destacada, a metodologia deve ser melhor descrita e uma figura inserida.
Muitos parágrafos sem referências ao longo do texto. Mesmo que o parágrafo se refira ao assunto discutido no parágrafo anterior, a referência deve ser citada:
Linhas nas quais as referências devem ser inseridas:
118, 176, 182, 281, 346, 356, 370, 392, 404, 410, 414, 432, 442.452.465.472, 474, 486, 500, 509, 518, 535, 539, 600, 606, 638, 645, 659, 672, 684, 718, 725, 786, 793, 870, 917 e 927.
Reviewer 2 Report
Comments and Suggestions for Authors
I recommend that the authors consider adding figures to enhance the visual appeal of the manuscript and engage readers more effectively. Including diagrams, graphs, and a graphical abstract could help clarify complex concepts and key findings. Additionally, the tables could be redesigned to improve readability, using color coding, clearer headings, and summary rows to highlight the most important data.
I also find that the manuscript does not clearly highlight what is unique about this review paper or what new insights it brings to the field, as there are already existing reviews on similar topics—raising the question of whether there is a distinct need for this review.
Abstract
Emphasize what makes this review unique—whether it's new encapsulation methods, recent advancements, or novel applications in the food industry. This helps readers quickly understand the paper's distinct contributions.
Briefly mention specific encapsulation techniques or wall materials covered (e.g., spray drying, coacervation). This adds clarity and helps readers anticipate the paper’s technical depth.
Reinforce that encapsulation offers benefits beyond stability, like controlled release or improved bioavailability, to underscore its full value for food applications.
Define the range of applications for encapsulated EO and OR in the food industry. This sets expectations and demonstrates the practical significance of the review’s findings.
Simplify complex sentences to improve readability and ensure a logical flow from problem statement to solution. This makes the abstract more accessible and engaging.
Introduction
Emphasize the shift toward natural and health-promoting ingredients, connecting it to consumer health concerns and regulatory pressures. This strengthens the rationale for using EO and OR over synthetic additives.
Expand on specific formulation challenges (e.g., volatility, degradation, and hydrophobicity) that hinder the use of EO and OR in foods. This grounds the need for microencapsulation in real-world industry applications.
Highlight how microencapsulation not only improves the stability and shelf life of EO and OR but also masks undesirable flavors and odors, showcasing its multifunctional benefits.
Provide specific examples of how microencapsulation is used in food products (e.g., flavored beverages, shelf-stable snacks). This makes the applications more tangible and relatable for readers.
Briefly conclude with how this review addresses research gaps or explores recent advancements in encapsulation techniques, underscoring the paper’s relevance and contribution.
Microencapsulation techniques
Clearly distinguish the three categories of microencapsulation methods. For example, add a sentence explaining how physical, chemical, and physicochemical methods differ in terms of energy, interaction with core materials, and applications in food science. This sets up readers with a stronger understanding of why a particular method may be chosen based on the core material properties and intended functionality.
Emphasize practical aspects that impact industry adoption, such as production costs, efficiency, and environmental implications of these techniques. You could mention any regulatory or logistical considerations that make some techniques more favorable at an industrial scale than others. A sentence on factors like sustainability or compatibility with existing food processing equipment would add relevance to the discussion.
When discussing emerging encapsulation technologies, briefly mention why these new methods were developed. For instance, explain that they aim to address limitations in stability, scalability, or environmental impact compared to traditional methods. This transition can bridge the gap between the well-established techniques and the innovation-driven approaches described later.
Consider reformatting Table 2 to highlight each technology’s advantages and limitations, making it easier for readers to compare options. Group similar techniques, like variations of spray drying, for clarity. Adding a column or visual cue (like color-coding) for key benefits in commercial scalability or bioavailability might also make the table more user-friendly.
Expand on the discussion of scaling challenges by referencing a few industry examples where successful upscaling has been achieved (e.g., spray drying’s widespread use) or remains challenging (e.g., nano spray drying). Providing tangible industry examples can give readers insight into the gap between lab-scale successes and commercial realities.
Conclude the section by identifying areas where further research could support large-scale implementation. This could include potential for cost reductions, development of eco-friendly wall materials, or alternative methods to increase encapsulation efficiency and stability under real-world storage conditions.
Encapsulation Materials
Add specific examples of how different release triggers (e.g., pH-sensitive or heat-sensitive coatings) impact the choice of coating materials, making the material selection process clearer and more practical.
Elaborate on how antioxidants, surfactants, and emulsifiers improve microcapsule stability and functionality (e.g., antioxidants for shelf-life, surfactants for better dispersion), giving readers a stronger understanding of additive functions.
Include a table listing key properties, benefits, and limitations of frequently used encapsulation materials (e.g., starch, proteins, gums), which would serve as a quick and valuable reference.
Core Materials - Essential Oils and Oleoresins
Start this section with a concise comparison between EOs and ORs, particularly focusing on stability and compositional differences, such as the resin component in ORs offering better heat stability.
Expand on how encapsulation aids in dispersing EOs in aqueous environments, addressing solubility issues directly, and highlighting any agents or methods effective for liquid-based foods.
Food and beverages applications
Consider highlighting the advantages and limitations of specific wall materials for different encapsulation processes, such as their impact on flavor release, retention, and stability. This provides insight into choosing the most suitable materials for specific applications.
Expand on the influence of storage conditions such as temperature, humidity, and light exposure on the stability of encapsulated flavors. Exploring this will help deepen understanding of how to maintain flavor quality over time.
Discuss the role of antioxidant properties in encapsulated flavors, particularly in terms of how they contribute to both preserving the flavor and enhancing shelf-life in food products.
Future perspectives and conclusions
Emphasize the need for further research into affordable microencapsulation technologies to make EO and OR more accessible and sustainable for the food industry.
Stress the importance of finding alternative, cost-effective wall materials with similar properties to traditional ones, to improve the feasibility of large-scale microencapsulation in food applications.
